# Preliminary Determination of the Optimal Parameters When Using an Ultrasonic Probe to Measure Cavern Geometry Where a Metal Borehole Pipe Is Present

**Tomasz Kubacka [1,2] and Chau Nguyen Dinh [1,*]**

1 Department Geophysics, Faculty of Geology, Geophysics and Environmental Protection, AGH University of Science and Technology, al. Mickiewicza 30, 30-059 Kraków, Poland; tomaszkubacka@agh.edu.pl or tomasz.kubacka@chemkop.pl

2 The Research and Development Centre for Mining of Chemical Raw Materials, CHEMKOP Sp. z o.o., ul. Józefa Wybickiego 7, 31-261 Kraków, Poland

* Correspondence: cnd@agh.edu.pl; Tel.: +48-12-617-39-83

**Abstract:** In order to determine the optimal parameters when using an ultrasonic probe to measure cavern geometry when a metal borehole pipe is present, an investigation was firstly carried out on influence of a vertical metal plates with a thickness from 1 mm to 15 mm immersed in water on transmitted and reflected ultrasonic waves. The results obtained will be used as an indicator for the measurement of underground geometry in which the ultrasonic probe is placed inside a metal pipe lining a borehole. These studies were performed both by experiment and computer simulation. The results show that the wavelength of the incident ultrasonic signals should be equal to half the thickness of the metal plate or an integer times smaller than this thickness. When the thickness of the barrier is unknown, an ultrasonic signal with linear frequency modulation (LFM) should be used. Due to the reverberation of the ultrasonic waves inside the pipe for caverns filled with water, the distance from the transducer to the cavern wall can be measured if it is longer than three times of the pipe diameter. Frequency analysis of both the reflected and the transmitted waves enables an optimal frequency of the incident ultrasonic wave to be selected, which can be used in the measurement of cavern geometry in conditions in which the ultrasonic probe is inside a metal pipe.

**Keywords:** ultrasonic wave; acoustic impedance; steel plate barrier; salt cavern





## 1. Physical Background

Caverns formed after the leaching of salt domes are increasingly used as storage tanks for hydrocarbon fuel in a liquid or gaseous state or for compressed air as well as for hydrogen [1–7]. It is also common to use salt caverns for $CO_2$ sequestration [8] and waste storage [9]. Determining their shape and geometry is therefore of key importance for the proper and controlled performance of the processes involved in exploiting the deposit and subsequent long-term storage. For distance measurement there are several methods such as light optic sensor, georadar, and ultrasonic methods, but both georadar and light optic sensors in ground caverns are totally not successful, therefore in the case currently the best methods for the determining of cavern geometry is the ultrasonic wave method [10,11]. The theory and application of the ultrasonic methods have been known for several dozen years [12–14]. In our study case, echometric measurement consists of multiple recordings of a signal sent from an ultrasonic probe and the signal reflected from the cavern wall [7,15]. The distance from the sonar to the cavern wall can be calculated by formulae $d = \frac{vt}{2}$ where : $v$ speed of the ultrasonic wave $\left[\frac{m}{s}\right]$ and $t$—time between the moment of sending the signal and of recording the reflected signal [s].

Such measurement is usually carried out in caverns where there is an absence of operational pipes. However, in practice there are many cases where pipes remain in

caverns. Such situations include caverns used for the storing liquid hydrocarbons and which are accessible through a single borehole (Figure 1).

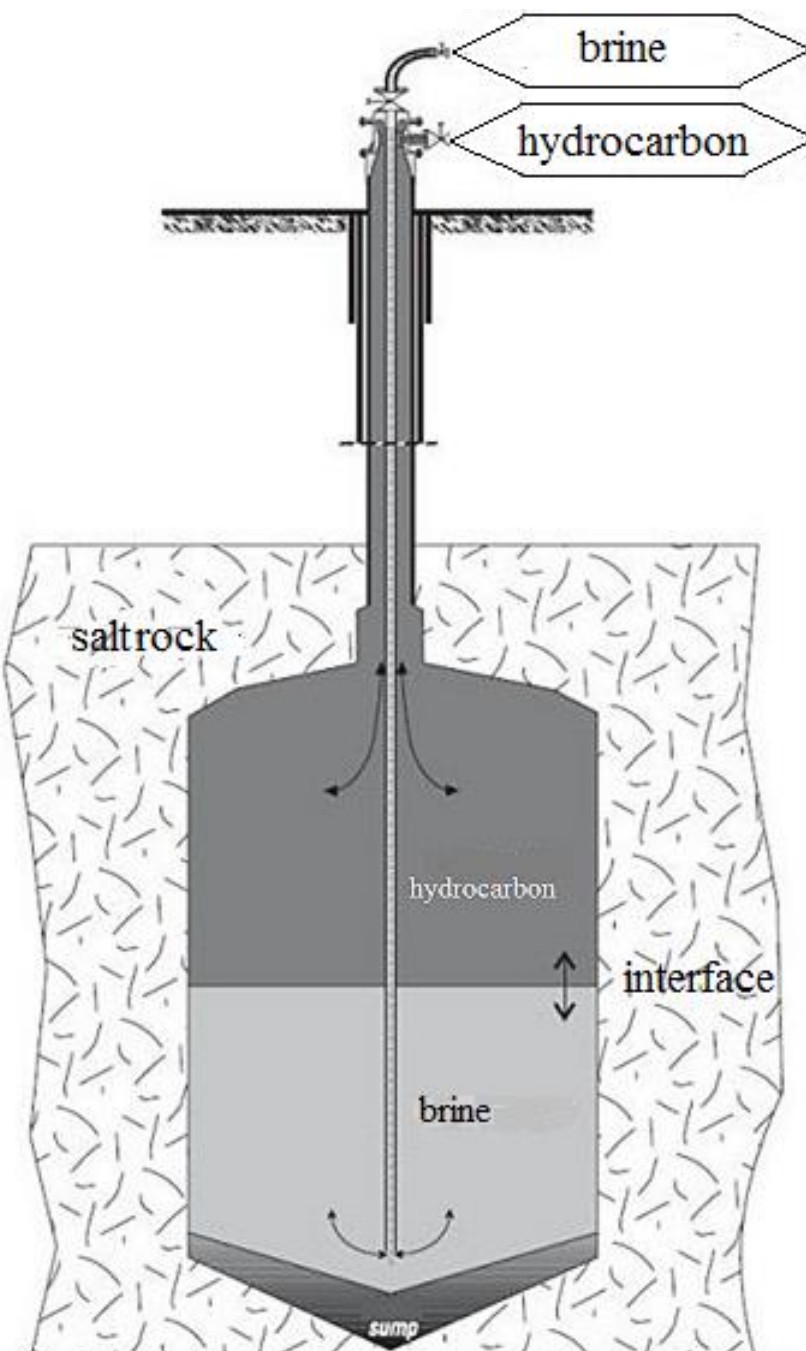

**Figure 1.** Schematic diagram of a cavern used as a container for liquid fuel with a steel pipe modified from [16].

Sometimes the service pipes are left in the caverns for economic reasons or due to technical problems associated with attempting to pull the pipe out of the borehole [17]. In such cases the measurements of the shape and geometry of the caverns must be performed using an ultrasonic probe placed inside the operating pipe. In industry such pipes are mostly steel pipes with diameters of $8^5/_8$" (219.08 mm) or $7^5/_8$" (193.68 mm) with a wall thickness of approximately 10 mm and pipes that are 5" (127 mm) or $4^1/_2$" (114.3 mm) in diameter with a wall thickness of approximately 5–6 mm. Occasionally, pipes $10^3/_4$"(273.05 mm),

$9^5/_8$"(244.48 mm), $6^5/_8$" (168.28 mm), and $5^1/_2$"(139.70 mm) in diameter are also used. The presence of the steel pipes in air or a liquid medium significantly affects the transmission of the ultrasonic signal; it also affects the orientation of the ultrasonic probe in the borehole. Using an ultrasonic probe in this situation, a down-hole instrument sensor in an appropriate position for the probe is used [16]. This situation causes enormous difficulties in carrying out an accurate and reliable measurement. In addition, there is a change in the wall thickness of the service pipes resulting from corrosion or the accumulation of deposit on their surface. Such changes are usually troublesome when determining an appropriate frequency for the ultrasonic signal which is least weakened by a barrier in the way of its passage to the wall of the cavern being measured and then recorded using the receiver in the probe. If a proper frequency is not chosen, this causes a major weakening of the signal amplitude and thus limits the range and precision of the measurement. Generating a wider signal frequency band also does not solve the problem due to reverberation inside the pipe. The signal reflected from the inside of the pipes will be superimposed on the echo signal record, making it difficult to interpret the results. Therefore, it is important to find a way to select an ultrasonic frequency of signal suitable for the specific situation of measurement.

In practice, measurement procedure of a cavern geometry relays on that the echo probe is gradually lowered from top to bottom in the borehole and at a given point the probe measures distance from it to the point at the cavern wall, this point and the transducer center are on the horizontal line. To draw a horizontal circumference of the cavern wall the echo probe is shifted per a given angle (usually 1° to 5°) around the vertical axis. At every angle, the distance to the wall is measured.

A great difference in acoustic impedance $\Delta Z (\Delta Z = Z_1 - Z_2$, where: $z_1 = \rho_1 \cdot v_1$, $z_2 = \rho_2 \cdot v_2$; $z$—acoustic impedance, $\rho$—medium density [kg/m³], $v$—speed of ultrasonic wave) between media, causes a strong reflection and weak penetration of the ultrasonic wave at the interface between the media.

At a boundary of two media and according to the principal continuity and assuming that there is no loss energy for dispersion and for heat generation in wave propagation, the incidence, reflection, and transmission pressures and speeds can be expressed by the equations as follows:

$$P_i + P_r = P_t \tag{1}$$

$$v_i + v_r = v_t \tag{2}$$

where $P_i$, $P_r$, $P_t$, and $v_i$, $v_r$, and $v_t$ are the pressure and speed of incidence, reflection, and transmission waves, respectively.

In practice the pressure of reflection and transmission waves are usually characterized by the reflection ($R_d$) and transmission ($D_d$) coefficients defined as the ratio of the pressure of reflected wave and transmitted wave to the pressure of incidence wave respectively. In the case of the plane wave propagating perpendicular to the plate (Figure 2), the reflection and transmission coefficients $R_d$ and $D_d$ through a thin, flat vertical plate with impedance $Z_2$ immersed in media with and $Z_1$ can be calculated by the formulas [16,18]:

$$R_d = \frac{\left| \frac{1}{4} \left( \frac{Z_2}{Z_1} - \frac{Z_1}{Z_2} \right)^2 sin^2 \left( \frac{2\pi g}{\lambda} \right) \right|}{\left[ 1 + \left( \frac{1}{4} \left( \frac{Z_2}{Z_1} - \frac{Z_1}{Z_2} \right)^2 sin^2 \left( \frac{2\pi g}{\lambda} \right) \right) \right]} \tag{3}$$

$$D_d = \frac{1}{\left[ 1 + (\frac{1}{4} \left( \frac{Z_2}{Z_1} - \frac{Z_1}{Z_2} \right)^2 sin^2 \left( \frac{2\pi g}{\lambda} \right) \right]} \tag{4}$$

where: $g$ is the thickness of the plate [m], $\lambda$ is the wavelength of the longitudinal wave propagating in the plate [m].

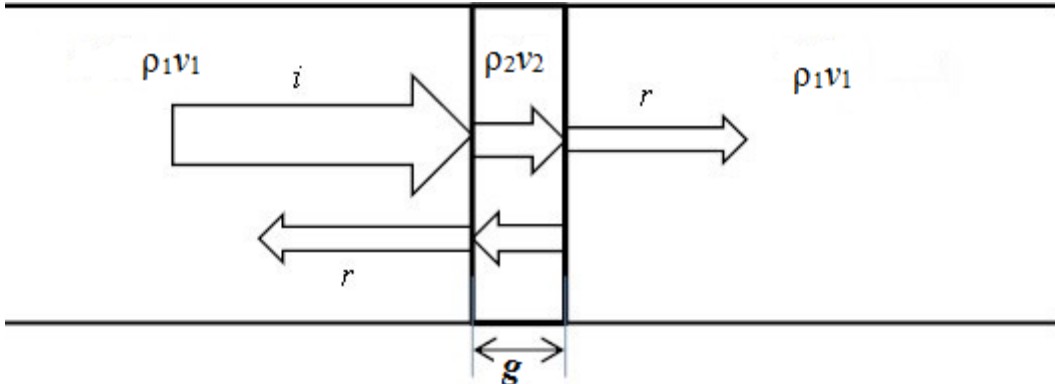

**Figure 2.** Schematic diagram showing the movement of an ultrasonic wave through a barrier plate with thickness g and impedance $\rho_2 \, v_2$ within a medium with impedances $\rho_1 \, v_1$.

Based on the data obtained from the calculation using Formulas (2) and (3), a graph was drawn of the dependence of the transmission and reflection coefficients of the ultrasonic waves from the vertical steel plate on the ratio of the wavelength to the thickness of the steel plate immersed in water (Figure 3).

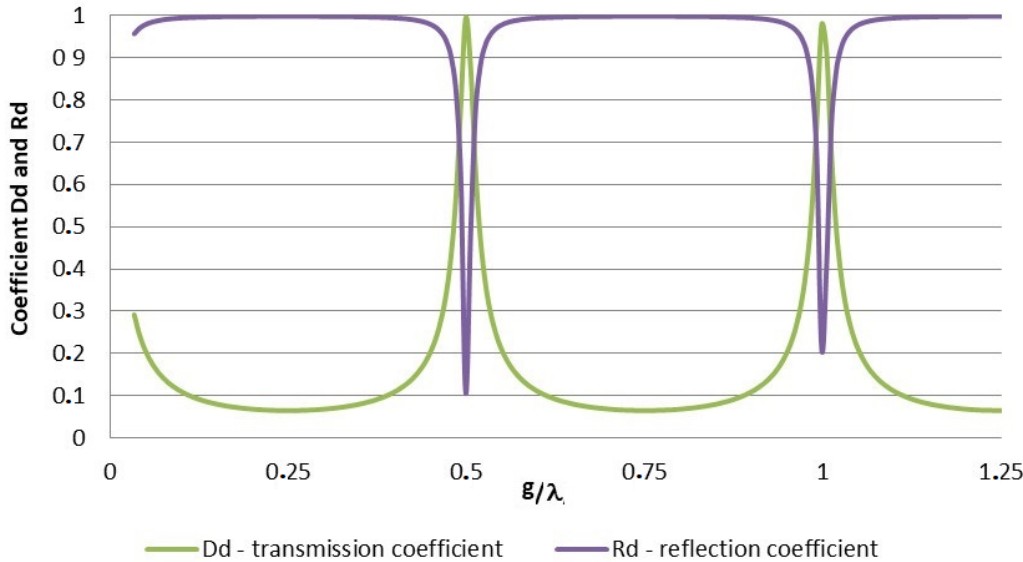

**Figure 3.** Dependence of the transmission $D_d$ and reflection $R_d$ of an ultrasonic wave on the ratio of the wavelength and the thickness of the steel plate barrier.

Figure 3 shows that the transmission and reflection coefficients of the wave have extreme values at points where the wavelength is a multiple of half of the plate thickness. This observation has also been described by numerous scientists [19–21].

When the measurement is carried out, the direction of the wave should be perpendicular to the plane of the barrier [22,23]. When both the probe and barrier are in water, the barrier is coupled with aqueous medium and the transmission of the wave will be better [24].

In practice, there are some methods of modifying the incident pulse that can reduce the energy loss in the obstruction. For example, we can excite a transmitting transducer with single or multiple waves with a frequency in a resonant range (Figure 4a), or by increasing the echo energy band by energizing the transducer with a single pulse with a longer duration (Figure 4b). In ultrasound technology, technicians increasingly use a linearly modulated frequency domain (LFM) signal (Figure 4c). When exciting the transducer with an LFM pulse, it is not only the length but also the start and end frequencies that are to

be determined. The LFM signal also causes a significant increase of the inherent signal to noise ratio [25].

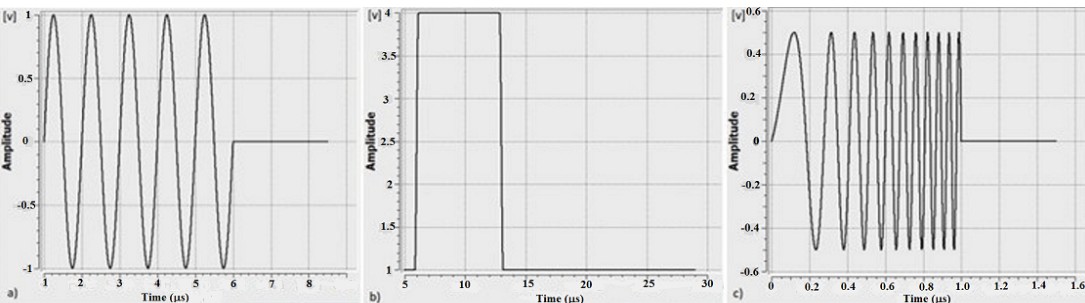

**Figure 4.** Types of incident signals: (**a**) signal with a resonant frequency (tone burst); (**b**) square pulse signal; (**c**) signal with linearly variable frequency (LFM).

## 2. Experiments

### 2.1. Influence of the Thickness of the Steel Plate Barrier on the Ultrasonic Wave Transmitted

In this study three types of transducers were used, first is piezoelectric with 25 mm of diameter produced by the Chemkop Company®, Krakow, Poland, its working frequency and beam angle are 25 kHz and 95°, 70 kHz and 30°, and 100 kHz and 21°. The second is also piezoelectric, but with 38 mm diameter produced by the Echologger Company®, Gyungki-Do, Korea, its working frequency and beam angle are 200 kHz and 10° and 450 kHz and 5°. The third is magnetostrictive transducer in a cuboid with 55 mm × 55 mm × 100 mm produced by Chemkop Company®, its working frequency 47 kHz and a space angle with 25° of azimuth and 13° of altitude. The ultrasonic wave pressure of the all mentioned transducers is 3 dB. The hydrophone receiver is a product of the Brüel & Kjær 8103™ Company, Nærum, Denmark and its receiving sensitivity is 211 dB re 1 V/µPa. The detailed description of the mentioned transducers and receiver can be downloaded from the companies' web pages [26–28].

The research on the influence of the plate barrier on the transmission signal was carried out on a special stand (Figure 5). This stand consists of a rectangular container with dimensions of 2 m × 1.8 m × 1 m filled with tap water to a height of 0.8 m. A hydrophone receiver (Rx) was placed in it at a distance of 1.5 m from the transducer. Ultrasonic signals recorded by Rx were amplified by a linear amplifier. The ultrasonic transmitting signal Tx was generated by piezoceramic ultrasonic transducers with frequencies of 25 kHz, 70 kHz, 100 kHz, 200 kHz, 450 kHz, and from a magnetostrictive transducer with a frequency of 47 kHz.

Several plates with thickness equal to 2, 4, 6, 8, 10, 12, and 15 mm, respectively, were placed at a point 400 mm from the transducer on the line between it and the hydrophone. The plates were all the same length (500 mm) and width (400 mm). The incident signal in the form of a single pulse or group of three sinusoidal pulses was triggered from a Valleman function generator PCGU1000™ and amplified by a power amplifier. The signal received from the hydrophone was amplified and displayed on a Valleman PCSU1000 digital oscilloscope. For reference purposes, at the very beginning signals were measured in the same stand but with no steel plate, then the amplitude of the signal was measured after passing through the steel plate. The influence of the distance l between the plate barrier and transmitting transducer on the signal recorded was also investigated. The water temperature was constant and equal to 20 °C, the velocity of the ultrasonic wave in water was measured with a Valeport Mini SVS device and was 1482.9 m/s. The acoustic characteristics of the water and steel used in the laboratory are summarized in Table 1.

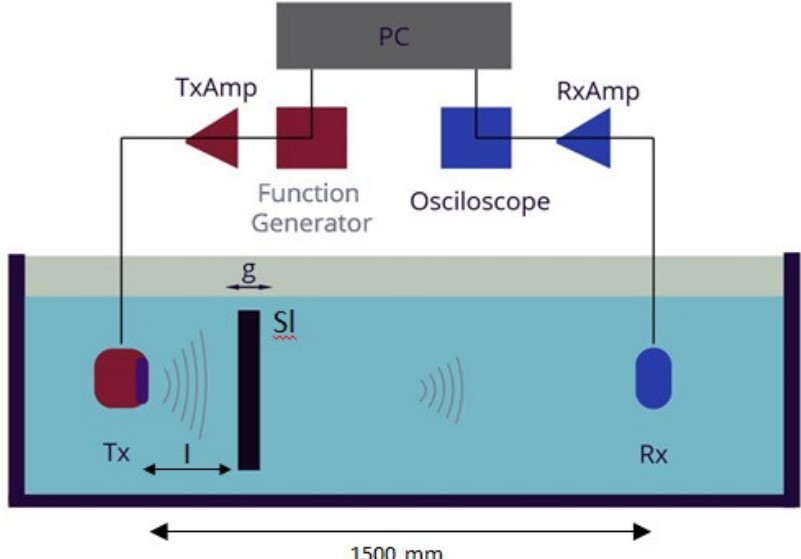

**Figure 5.** Schematic diagram of the stand for researching the transmission of an ultrasonic wave through a steel plate barrier; Tx—ultrasonic transducer; Rx—receiver (hydrophone); Sl—steel plate with thickness g; l—distance between the plate and the transducer.

**Table 1.** Acoustic characteristics of the water and steel used in the laboratory.

| Parameter/Medium | Water | Steel |
|---|---|---|
| Density [kg·m$^{-3}$] | 1000 | 7800 |
| Wave speed [ms$^{-1}$] | 1483 | 5850 |
| Acoustic impedance [kgs$^{-1}$m$^{-2}$] | $1.48 \times 10^6$ | $45.63 \times 10^6$ |

*2.2. Influence of the Distance between the Transducer and the Plate Barrier on the Transmitted Signal*

The influence of the distance between transducer (Tx) and plate barrier was also investigated. This investigation relied on the measurement of the amplitude of the ultrasonic wave transmitted through the plate barrier placed at different distances from the transducer. The distance between the receiver (Rx) and the transducer (Tx) was 1500 mm and was not changed for the whole series of experimental readings. The experiments were done using ultrasonic waves with frequencies of 200 and 450 kHz for plate barriers with thicknesses of 2 mm, 4 mm, 8 mm, and 10 mm and for a distance between the transducer Tx and the plate barrier that was varied in the range from 2 cm to 50 cm.

**3. Computer Simulations**

*3.1. Transmission through the Barrier Plate and Reflection from Both Plate and Reflective Layer*

The next step was to carry out simulation studies on the influence of the steel plate on the received signal in a 2D model with 1000 mm × 300 mm using the OnScale™ program. The software of this program was written based on the finite element method. To obtain a high accuracy of the simulation, the model was divided on the square elemental mesh with a box equal to one tenth of an ultrasonic wavelength in simulation.

On the two-dimensional model there is an Tx ultrasonic wave generator and three receivers R1, R2, and R3, placed from Tx at distances of 130 mm, 470 mm, and 750 mm, respectively. The end right-hand edge of the model was the reflective boundary (Figure 6), while the three remaining boundaries were assumed to be wave-absorbing, i.e., not reflecting the signal (Figure 6). The first series of simulations was made with a steel plate with a thickness from 1 to 15 mm, variable in 1 mm steps placed at a distance 300 mm from the point of transmission. The signals transmitted were package of the sinusoidal waves with frequencies of 200 kHz and 450 kHz, both waves were induced by the same pressure with

5 Pa of amplitude, however the spatial emission angle is 6.3° for the first wave and 2.8° for the second one. The second series were simulations in the model in which LFM signals were reflected from and passed through plates 5 mm and 10 mm thick and were recorded by the R1, R2, and R3 receivers, respectively (Figure 6).

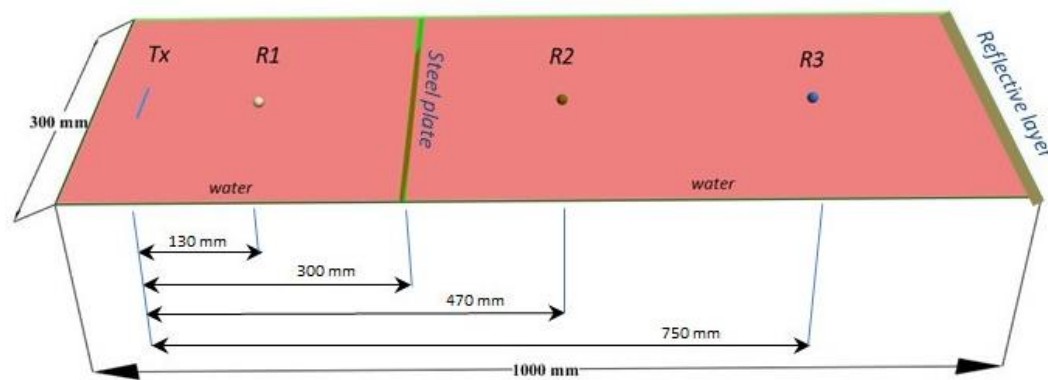

**Figure 6.** Simulation model 2D for researching ultrasonic waves passing through a steel plate barrier in water. Tx—transducer ultrasonic waves, R1, R2, R3—wave receivers.

The signals transmitted had linear frequency modulation in the ranges of 100–800 kHz and 550–650 kHz with 5 Pa of pressure amplitude. The frequency spectra of the signals received were also analyzed. The duration of every simulation was 1.5 ms.

### 3.2. Reverberation of Ultrasonic Waves in the Zone within the Plates

To investigate an influence of the reverberation effect on the recorded ultrasonic wave in the zone within the pipe, firstly we conducted a simulation of the ultrasonic wave behavior on a 2D model of a water aquarium 1000 mm × 300 mm. In the aquarium there were two steel plates with a thickness of 10 mm, and they were 219 mm apart (Figure 7). The thickness of the steel plates and the distance between them are well fit to the geometry parameters (thickness and diameter) of the standard pipes, which are most often used during the salt deposit exploitation. The transducer with a height of 50 mm emitted five times an ultrasonic wave with either a frequency of 290 kHz and 4.5° of beam width or a wave with 590 kHz and 2° of beam width. The ultrasonic transmitter Tx and receiver Rx were placed on the rectangle box side surface facing the reflecting edge of the model (Figure 7). The rectangle box is 100 mm high and 70 mm wide and composed of a specific mixture of epoxy and tungsten [29]. This material is characterized by 2975 $\text{kgm}^{-3}$ of density, 1960 $\text{ms}^{-1}$ of sonic wave speed and $4.2 \times 10^{-1}$ $\text{dB·mm}^{-1}\text{·MHz}^{-1}$ of the weakness coefficient. Such a material makes it possible to significantly attenuate the ultrasonic wave in the back plate. The simulation time was 1.5 ms.

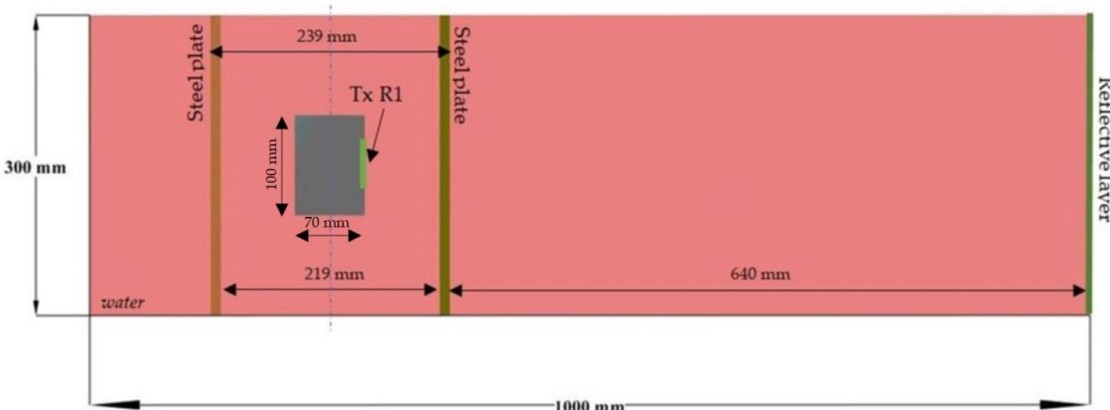

**Figure 7.** Model of a water aquarium with a transducer Tx and receiver Rx placed between two steel plates.

# 4. Results and Discussion

## 4.1. Experimental Results

4.1.1. Influence of the Thickness of the Plate Barrier on the Transmitted Signal

All the amplitude values of the recorded signals were compared to the amplitude value of the signals recorded when there was no steel plate barrier. The calculated values of the refraction part were obtained by subtraction of the transmission fraction from 100. Therefore, both transmission and reflection parts are expressed in percent. The relative amplitudes of transmitted signals are summarized in Table 2, and Figures 7 and 8 present the relative transmission and reflection fractions for the investigated frequencies of ultrasonic waves and thicknesses of the steel plates, respectively.

**Table 2.** The relative recorded amplitudes of the transmitted signals.

| Thickness | 2 mm | 4 mm | 6 mm | 8 mm | 10 mm | 12 mm | 15 mm |
|---|---|---|---|---|---|---|---|
| Frequency (Wavelength [a]) | Rx | Rx | Rx | Rx | Rx | Rx | Rx |
| 25 kHz (234 mm) | 65% | 50% | 35% | 22% | 25% | 25% | 21% |
| 47 kHz (124 mm) | 51% | 29% | 23% | 3% | 11% | 11% | 8% |
| 70 kHz (83.6 mm) | 37% | 17% | 25% | 3% | 6% | 6% | 4% |
| 100 kHz (58.5 mm) | 32% | 19% | 12% | 7% | 7% | 9% | 6% |
| 200 kHz (29.25 mm) | 19% | 10% | 8% | 4% | 9% | 12% | 29% |
| 450 kHz (13 mm) | 11% | 10% | 41% | 19% | 7% | 6% | 11% |

[a]—Calculated wavelength for the propagation of an ultrasonic wave in steel.

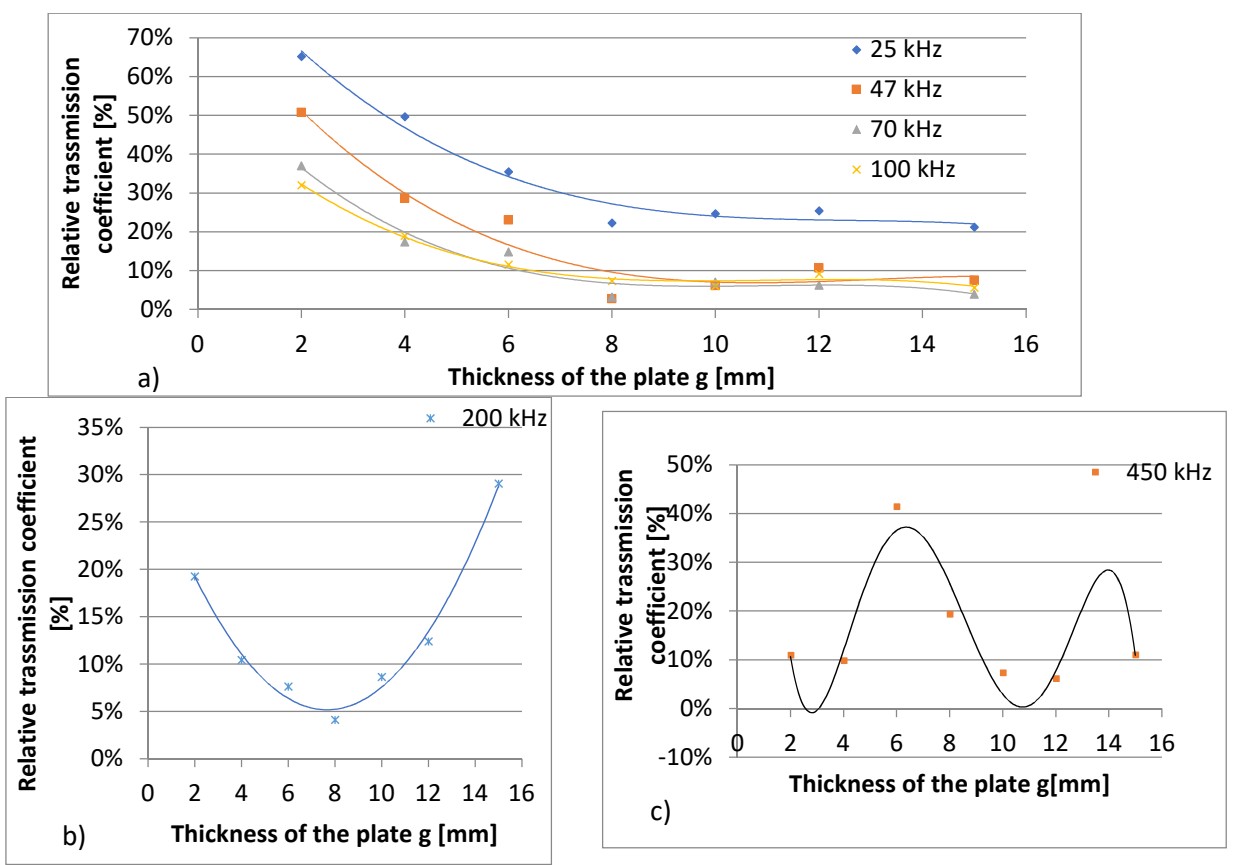

**Figure 8.** *Cont.*

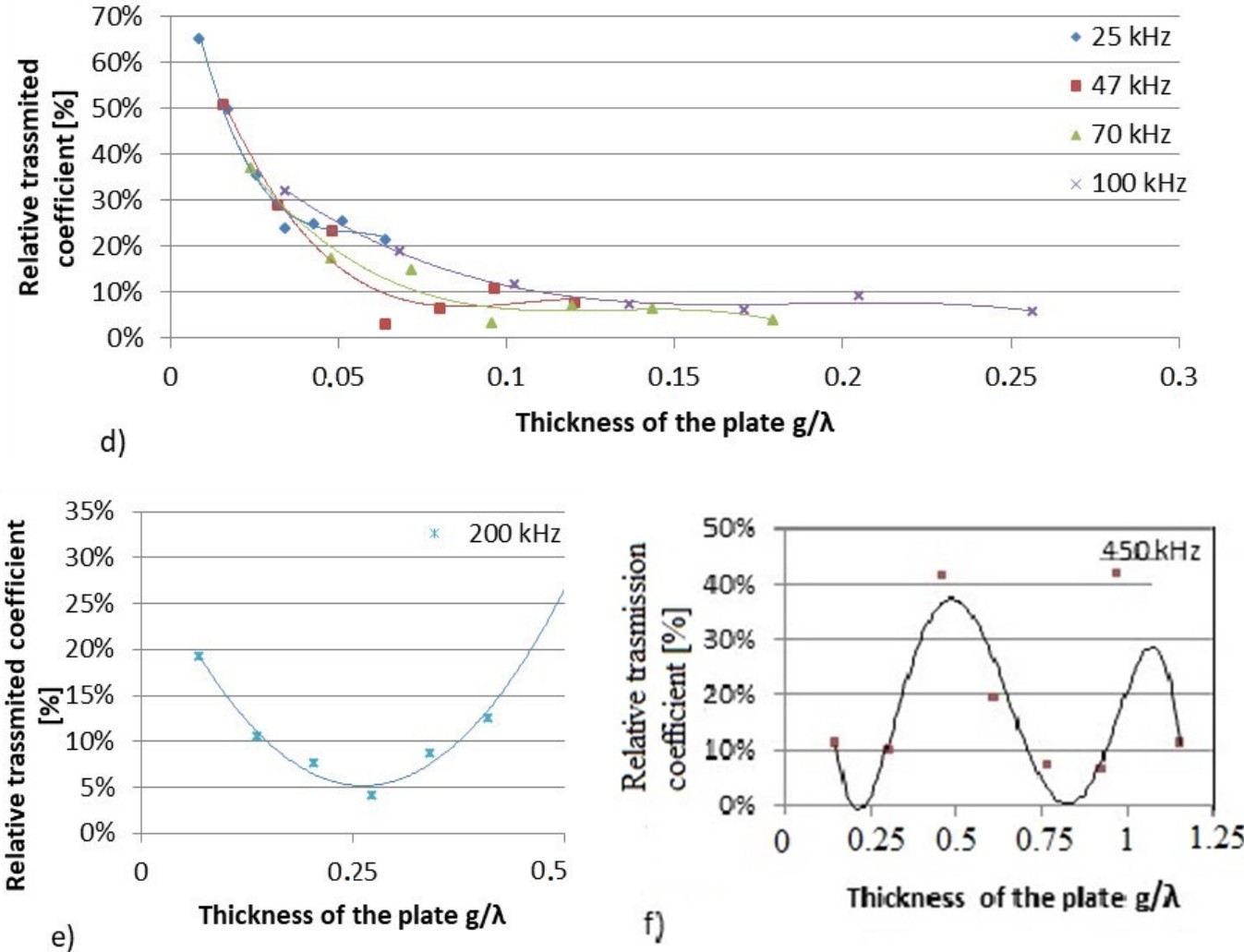

**Figure 8.** Relationship between the transmission coefficient and the thickness g of the steel plate: (**a**)—for ultrasonic waves with 25, 47, 70, and 100 kHz; (**b**)—for 200 kHz; (**c**)—for 450 kHz and plate thickness g expressed in [mm]; (**d**)—for 25, 47, 70, and 100 kHz; (**e**)—for 200 kHz and (**f**)—for 450 kHz and plate thickness expressed as the ratio [g/λ].

As can be seen from Table 2 and Figure 8a the transmission coefficient decreases with increase in both thickness (from 2 mm to 15 mm) and frequency (from 25 kHz to 100 kHz), and ranged from 65% at 25 kHz and a plate 2 mm thick to 6% at 100 kHz and a plate 15 mm thick. For the ultrasonic wave with 200 kHz of frequency the transmission coefficient decreases with the thickness of the plate and reaches 4% (minimum value) at the thickness of 8 mm, and then increases with increasing of the plate thickness (Figure 8b). For a signal with a frequency of 450 kHz the transmission coefficient reaches 41% (maximum) at a thickness of 6 mm (Figure 8c).

The behavior of the above-mentioned curves can be explained when these figures are presented in a system in which the plate thickness is expressed in units of the wavelength of the ultrasonic wave. Since the thicknesses of all the plates used in the experiments are lower than a quarter of the wave length of an ultrasonic wave of 25 kHz to 100 kHz in steel (Table 2), the transmission coefficients of all the above-mentioned waves decreases (Figure 8d).

For the ultrasonic wave with 200 kHz frequency, one quarter of its wavelength for steel is 7.31 mm (Table 2), so for a plate 8 mm thick the transmission coefficient is at a minimum (Figure 8e). For the wave with 450 kHz frequency, half of the wave length is 6.5 mm (Table 2), for a steel plate with thicknesses of 0.5 λ and 1 λ the transmission reaches a maximum and for thicknesses of 0.25 λ and 0.75 λ the transmission coefficient is at

a minimum (Figure 8f). Therefore, all the results obtained from the experiments are in agreement with the theoretical background.

### 4.1.2. The Influence of the Distance between the Plate Barrier and the Transducer Tx on the Transmitted Signal

Figure 9 presents the dependence of the measurements of the signals transmitted on the distance from the steel plate barrier to the transducer and shows that: 1. The amplitude of the signal transmitted decreases with increase in both plate thickness and wave frequency and 2. For a given frequency of wave emitted and a given thickness of plate barrier, the amplitude of the signal transmitted is stable and does not depend on the distance between the plate barrier and the ultrasonic transducer.

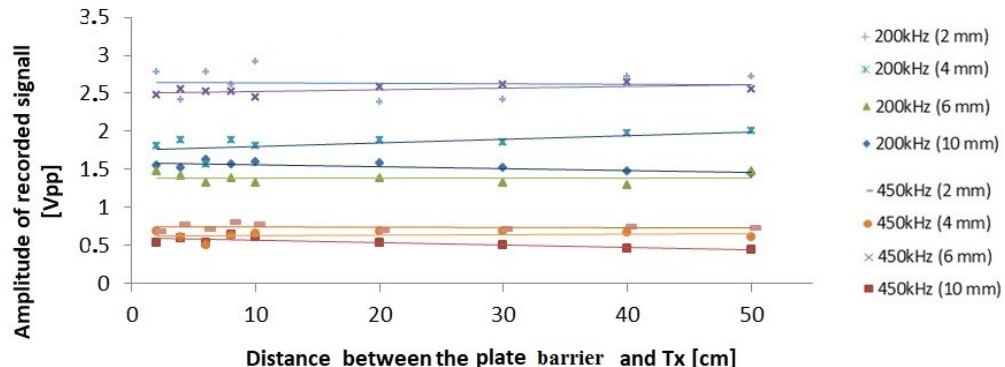

**Figure 9.** The influence of the distance l between the plate barrier and the transmitting transducer (Tx) on the recorded signal for different signal frequencies and plate barrier thicknesses.

### 4.2. Simulation Results

### 4.2.1. Influence of the Thickness of the Plate Barrier on the Transmitted Signal and Weakness Coefficient of the Plate Material

Figure 10a,b show the dependences of the transmission and reflection coefficients of ultrasonic waves with frequencies 200 kHz and 450 kHz on the thickness of the plate barrier expressed in unit of millimeters and in the ratio of $g/\lambda$, respectively. The extreme values of these coefficients are at the points, where the thickness of the steel plates is either half or one ultrasonic wave length.

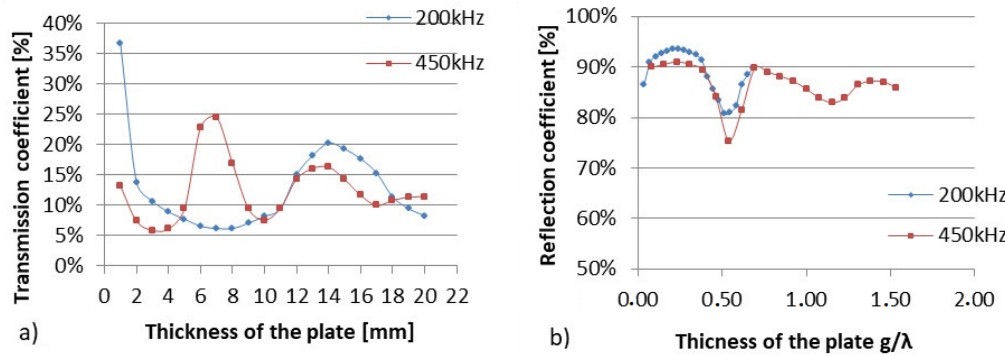

**Figure 10.** Dependence of the transmission (**a**) and reflection (**b**) coefficient on the thickness of the steel plate for ultrasonic waves with frequencies of 200 kHz and 450 kHz.

Figure 11a,b shows the relationships of the transmission and reflection coefficients obtained from the simulation for an ultrasonic wave with a frequency 100 kHz propagating in water, when there are steel plate barriers with different thicknesses g placed between the transducer (Tx) and receiver (Rx).

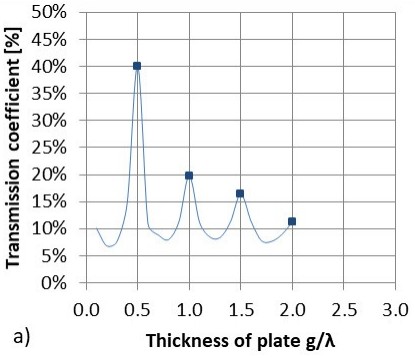
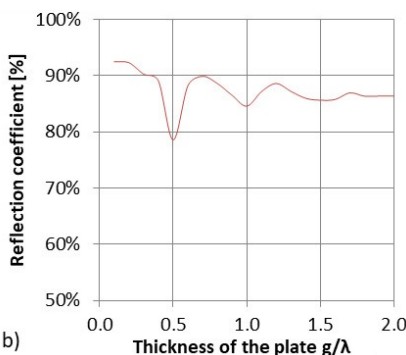

**Figure 11.** Relationship of the transmission (**a**) and reflection (**b**) coefficients on the thickness of the steel plate in water for an ultrasonic wave with a frequency of 100 kHz.

On the basis of the simulation results, the function expressing the dependence of the attenuation by a steel plate with different thicknesses g on an ultrasonic wave with frequency 100 kHz is as follows:

$$A = A_0 e^{-bx} \tag{5}$$

where $x$ is the thickness of the barrier plate expressed in units of the wavelength corresponding to 100 kHz in steel, $A_0$—amplitude of the ultrasonic wave without weakening, $b$—weakness coefficient and equal to $1.5 \pm 0.2 \, \text{Np}\lambda^{-1} \cdot 0.1 \, \text{MHz}^{-1}$. Taking into account the $\lambda$ corresponding to 0.1 MHz of ultrasonic wave in steel and the conversion factor from unit Np to dB, the estimated weakness coefficient is $2.8 \times 10^{-3} \pm 0.4 \, \text{dB} \cdot \text{mm}^{-1} \cdot 0.1 \, \text{MHz}^{-1}$. The table value of the weakness coefficient for steel ranges from $5 \times 10^{-3}$ to $5 \times 10^{-2} \, \text{dB mm}^{-1} \cdot \text{MHz}^{-1}$ [20] suggests that the estimated weakness $2.8 \times 10^{-3} \pm 0.4 \, \text{dB mm}^{-1} \cdot 0.1 \, \text{MHz}^{-1}$ can be accepted.

Both the experimental and the simulation results establish the dependence of the appropriate ultrasonic wave frequency, i.e., the one producing the maximum values for transmission on the thickness of the plate barrier, and this dependence is presented in Figure 12.

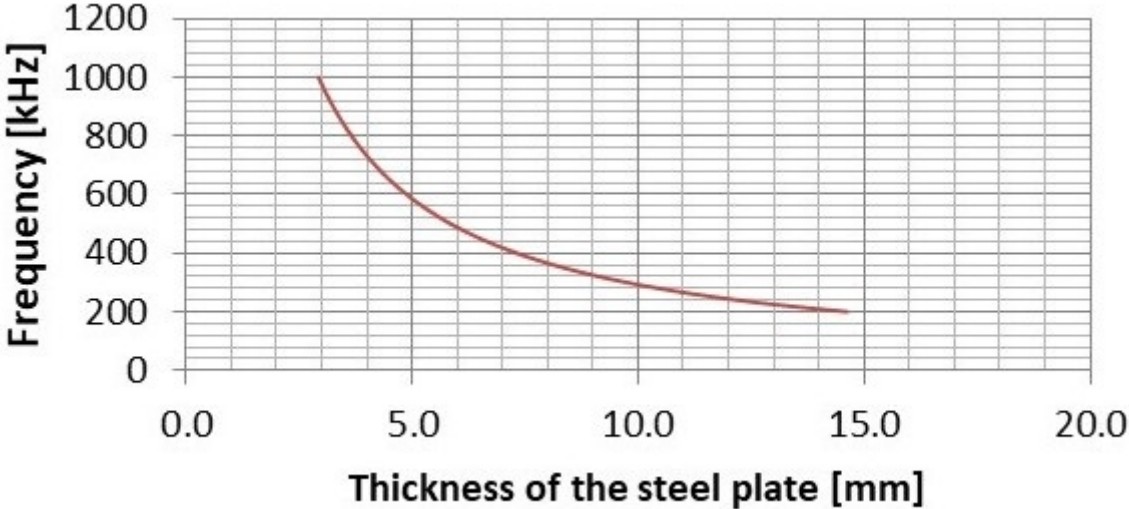

**Figure 12.** Dependence of the ultrasonic wave frequency at which the transmission has a maximum value on the thickness of the steel plate.

If there is an unknown thickness of the barrier or the thickness changes due to corrosion or sediment build-up, wave penetration tests can be performed using a signal with linear frequency modulation (LFM) (Figure 13a). In such a situation, signals with an appropriately wide frequency band are sent from the transmitting source Tx. The transmitted signals are recorded by receiver R3 placed on the opposite side of the barrier (Figure 13b).

Fourier analysis of the frequency characteristics (Figure 13c) of the recorded signals enables us to find the optimal frequency of the incident wave emitted from the source (Figure 13d).

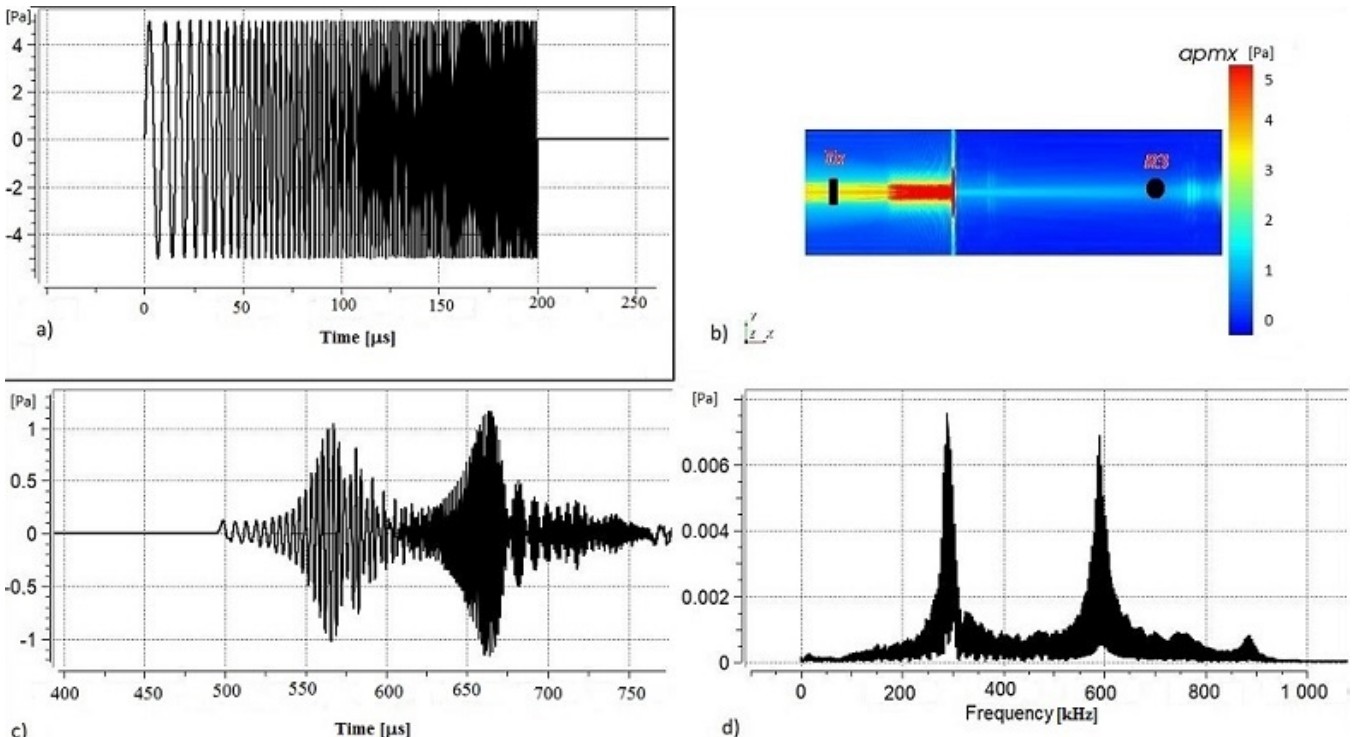

**Figure 13.** Results of the simulation of the transmission of an ultrasonic wave through a plate barrier with 10 mm thickness Figure 100. kHz LFM signal. Transmitted signals were recorded by receiver R3; (**a**) incident signal; (**b**) simulation of the maximum recorded signal; (**c**) signal recorded by receiver R3; (**d**) amplitude spectra of the signals recorded by receiver R3.

### 4.2.2. Transducer and Receiver Placed on One Side of the Barrier

In actual practice, we are dealing with a situation where we do not have a receiver located behind the barrier. This situation takes place in the echometric measurements of caverns in which we only have a device inside the exploitation pipes.

Referring to the above problem, simulation tests were carried out where both the emitter and receiver of the ultrasonic signals are placed on one side of the barrier. Simulations were performed for plates with thicknesses of 5 mm (Figure 14) and 10 mm (Figure 15). Receiver R1 (Figures 14b and 15b) was used to record (a) the modulated incident signals with frequency 550–650 kHz; (b) signals reflected from a barrier; (c) signals transmitted through the barrier and (d) signals reflected from the model boundary. The signals mentioned in cases of plates with thicknesses of 5 mm and 10 mm are presented in Figures 14a and 15a respectively. The reflected and transmitted signals recorded were subjected to Fourier transform analysis (FFT) and are presented in Figures 14c and 15c.

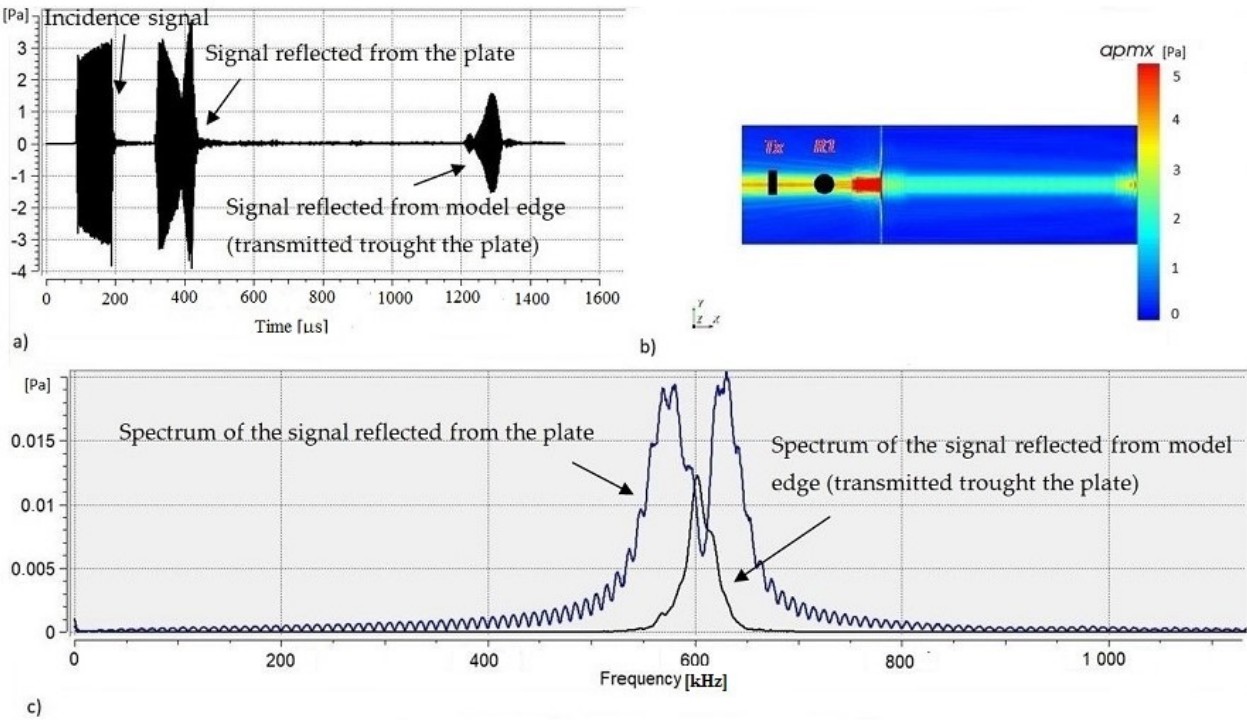

**Figure 14.** The transmission and reflection signals from transmitted and reflected ultrasonic waves in a water medium with 5 mm thick steel plate, incidental signal LFM 550–650 kHz; (**a**) recorded signal types; (**b**) pressure power of the signals inside the model; (**c**) the recorded spectra of the wave reflected from barrier plate (blue) and reflected from the reflect edge of model and transmitted again through the barrier plate (black).

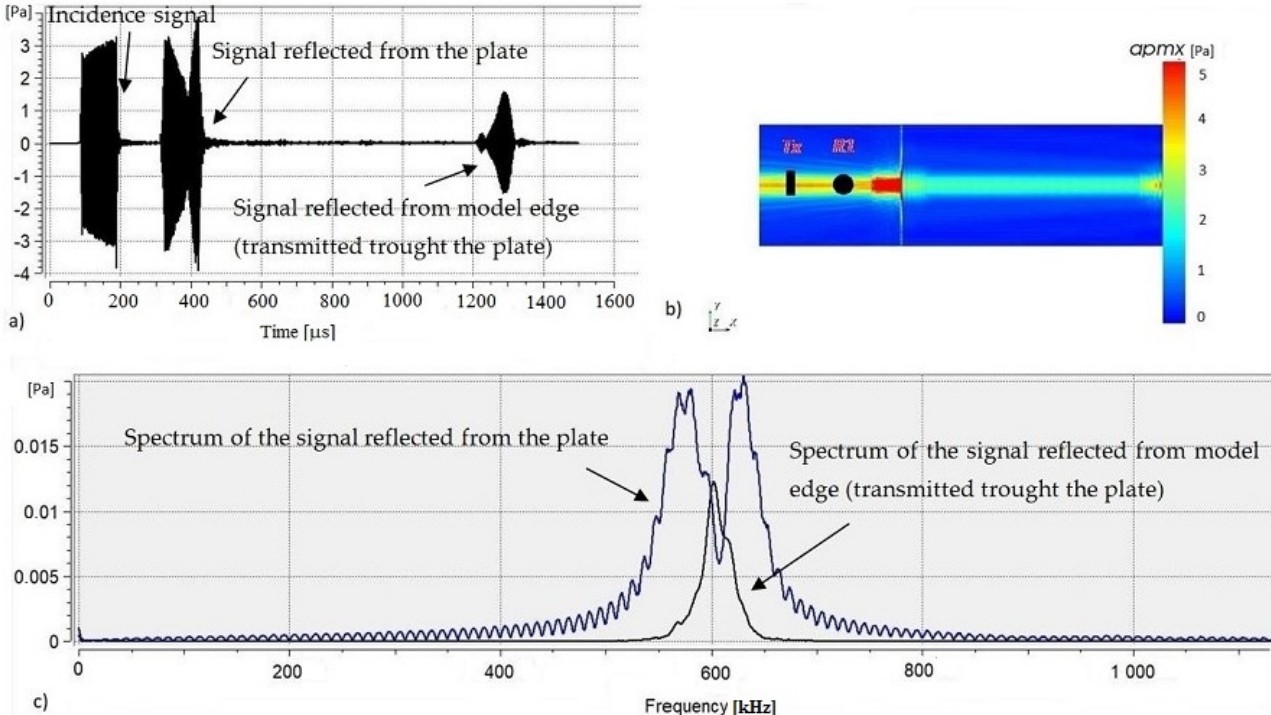

**Figure 15.** The transmission and reflection signals from transmitted and reflected ultrasonic waves in a water medium with 10 mm thick steel plate, incident signal LFM 550–650 kHz; (**a**) recorded signal types; (**b**) pressure power of the signals inside the model; (**c**) the recorded spectra of the wave reflected from the plate barrier (black) and reflected from the reflect edge of model and transmitted again through the barrier plate (blue).

Based on the analysis of the frequency characteristics obtained, an appropriate wave frequency can be selected to minimize the signal energy loss caused by reflection. For example, if an LFM signal ranges from 550 to 650 kHz and a barrier is 5 mm thick, one should select the signal frequency of 600 kHz (Figure 14c). However, for a thickness of 10 mm, the optimal frequency would be 590 kHz (Figure 15c). Figures 14c and 15c again indicate that when the thickness of the barrier plate is about half the wavelength, the loss of signal energy after transmission through the layer is smaller than in the case of a barrier with a thickness equal to the wavelength. The relative standard deviation ($\varepsilon$) of the determined steel barrier plate thickness depends on the relative ratio of the half width of the peak to the frequency at the peak center. In our case, the $\varepsilon$ is near 7%.

### 4.2.3. Transducer and Receiver Placed in the Zone between the Two Barrier Plates

The results obtained from the simulation processes for the incidence waves of 290 kHz and 590 kHz for the case where the Tx and Rx were placed between two steel plates with 10 mm of thickness are shown on the Figures 16 and 17, respectively.

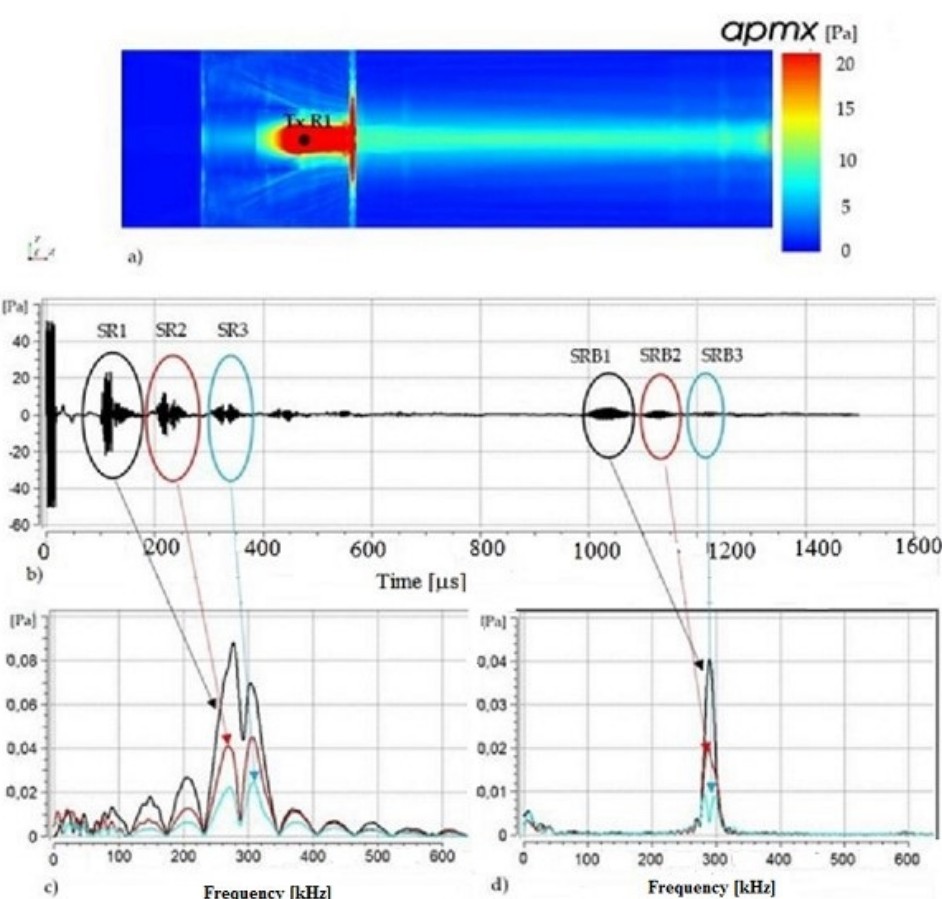

**Figure 16.** Results of the simulation for the case where the transducer and receiver are placed in the zone between two steel plates with 10 mm of thickness and frequency 290 kHz of incidence ultrasonic wave; (**a**) distribution of the wave pressure in the model; (**b**) recorded signals formed from reverberation in the zone between the plates (SR1, SR2, SR3) and from model reflection edge (SRB1, SRB2, SRB3); (**c**) the harmonic wave components obtained from Fourier analysis of SR1, SR2, SR3; (**d**) the harmonic wave components obtained from Fourier analysis of SRB1, SRB2, SRB3.

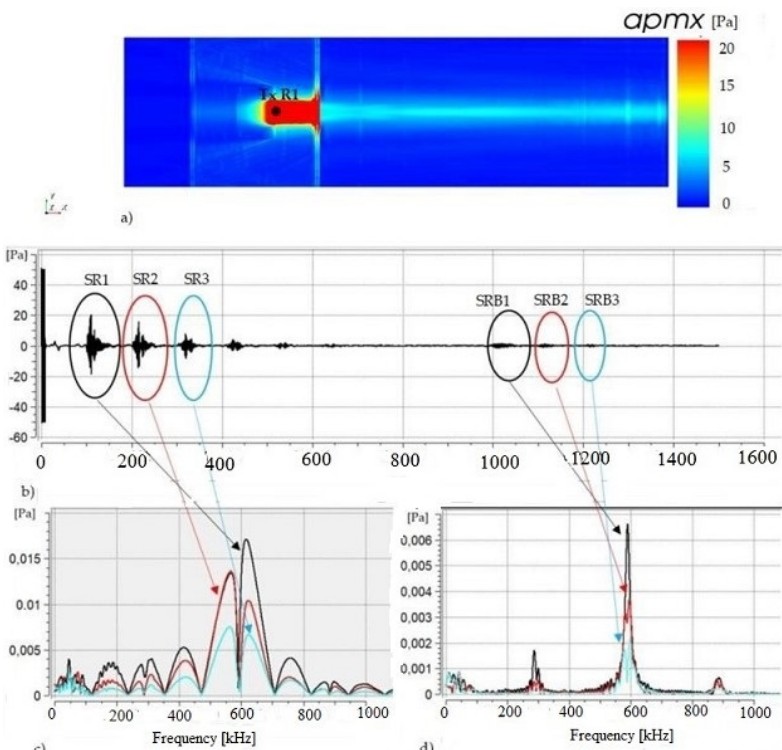

**Figure 17.** Results of the simulation for the case where the transducer and receiver are placed in the zone between two steel plates with 10 mm of thickness, and frequency 590 kHz of incidence ultrasonic wave; (**a**) distribution of the wave pressure in the model; (**b**) recorded signals formed from reverberation in the zone between the plates (SR1, SR2, SR3) and from model reflection edge (SRB1, SRB2, SRB3); (**c**) the harmonic wave components obtained from Fourier analysis of SR1, SR2, SR3; (**d**) the harmonic wave components obtained from Fourier analysis of SRB1, SRB2, SRB3.

The ultrasonic wave pressure is the most on the horizontal bank covering the transducer Tx (Figures 16a and 17a). Due to the presence of the weakening material in the transducer box, the wave pressure on the back of transducer is clearly weaker than those on the front of Tx (Figures 16a and 17a). Figures 16b and 17b are showing the recorded signals resulted from the first, second and third reflection (SR1, SR2, SR3) between the two plates and their corresponding signals SRB1, SRB2, and SRB3, which resulted from the reflection from the model edge, respectively. The time difference between adjacent recorded signals from either reverberation in the zone between two plates or refection from the model reflecting edge corresponds to twice the distance between two plates. Obviously, the recorded times of the SRBs signals are corresponding to the distance between Tx and the model reflecting edge after taking into account the number of reflections between two barrier plates. The harmonic wave components of the SR and SRB signals obtained from Fourier analysis are presented in Figures 16c and 17c. These figures indicate that (i) the waves with length of a half or equal to the thickness of the barrier plate have maximal amplitude; (ii) due to heat generation and dissipation during each incidence with plate the amplitude of the reverberation signals decrease rapidly, (iii) the distance from Tx to the reflecting cavern wall can be measure if it is at least equal to three times the distance between the plates (pipe diameter). The relative standard deviation ($\varepsilon$) is determined by the relative ratio of the length of the recorded signal reflected from the cavern wall (model reflecting edge) to the recording time of the recorded signal reflected from the cavern wall. In our case, the $\varepsilon$ is near 5%.

## 5. Conclusions

The results of the research carried out in this study indicate:

1.  Generally, the transmission coefficient of the ultrasonic wave decreases with increase in barrier thickness and wave frequency.
2.  The transmission coefficient reaches maximal values for a plate with a thickness equal to a multiple of half of the wavelength of the wave propagating in the barrier material. This conclusion confirms the results of the theoretical background.
3.  In the case where the distance from transducer and receiver is constant, the influence of the position of the plate in mentioned section on the transmitted signal recorded is not significant.
4.  Knowledge of the thickness of the plate barrier makes it possible to select an appropriate frequency for the signal to be emitted;
5.  In cases where the thickness of the plate barrier is unknown, the signal should be generated with a linearly modulated frequency LFM. The relative standard deviation is near 7%.
6.  Frequency analysis of the transmitted and reflected signals received makes it possible to determine the frequency of the signals which are least attenuated by the plate barrier.
7.  The reverberation of the ultrasonic wave between two barrier plates (pipe diameter) clearly influences on the recorded signals and in the case of cavern filled water the distance between the ultrasonic probe and the cavern wall must be at least longer than three times of the distance between the two places (pipe diameter). The relative standard deviation of the determined distance is near 5%.
8.  It is planned to extend the tests of both the transmitted and the reflected signals to cases where the measuring probe is inside single or double service pipes. Such cases are more and more frequently encountered in the exploration of salt domes.

**Author Contributions:** The conceptualization, methodology, software, validation, formal analysis, investigation, data curation, writing, review and editing were done by both the authors. Supervision by C.N.D., project administration by T.K. All authors have read and agreed to the published version of the manuscript.

**Funding:** This work is supported by UST-AGH Krakow, Grants no 16.16.140.315 and the Agreement on cooperation in the field of the Implementation Doctorate Programme.

**Institutional Review Board Statement:** Not applicable.

**Informed Consent Statement:** Not applicable.

**Data Availability Statement:** The data presented in this study are available in the authors' database.

**Conflicts of Interest:** The authors declare no conflict of interest.

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
