# Peer review of "Preliminary Determination of the Optimal Parameters When Using an Ultrasonic Probe to Measure Cavern Geometry Where a Metal Borehole Pipe Is Present"

_acoustics, doi:10.3390/acoustics3020028_

Round 1

Reviewer 1 Report

General appreciation of the paper:

The main objective of the paper is to study the ability of ultrasound to determine the geometry of a cavern through a metallic pipe. The introduction makes it difficult for the reviewer to appreciate the novelty of such a method in this specific context as almost all the papers cited on the subject of storage in caverns are in Polish. The authors should add more references published in international journals, and give details about the advantage of ultrasonic techniques versus other measurement techniques (for instance radar) in this particular application.

Besides, the paper presents experiments and simulations of a very simplified laboratory configuration, where the pipe is modelled as a plate immersed in water. The major concern of the reviewer is the fact that ultrasonic wave reflection and transmission in such a configuration is well known for decades, not only theoretically but also experimentally. Ultrasonic reflection and transmission in layered media has indeed been extensively studied, for instance for bonded structure inspection via spectroscopy (see for instance the works of Lavrentyev and Rokhlin, JASA 1997, followed by many others). Even though the experiments are well conducted, the obtained results are not new. Moreover, the effect of frequency dependence of ultrasound energy going through a layer is used on a daily basis in Non Destructive Testing when designing impedance adaptive layer for piezo-electric transducers ; the thickness and nature of the layer being chosen to enable maximum energy transmission of ultrasound in a medium at a given frequency.

However, the paper could be improved if the authors added experiments or simulations in a configuration more specific to cavern geometry measurement through a pipe.  Several parameters could affect the measurement, which have not been considered in other application fields, such as the curvature of the pipe, or the presence of hydrocarbons instead of water on one side, irregular cavern walls, and so on. At the very least, experiments in a configuration where the measuring probe is inside a pipe should be added to the paper.

Detailed comments:

Line 66 to 71: as stated by the authors, ‘the signal reflected from the inside of the pipes will be superimposed on the echo signal record’. However, the solution proposed (using LFM signal, line 262-263) would increase this difficulty, as LFM signals are longer in time than ‘the package of three sinusoidal waves’ (line 168) at a given frequency. Moreover, the statement line 66-68 is ambiguous, as it seems to imply that a signal with larger frequency band other than LFM would last longer than a signal with short frequency band, which is incorrect.

Figure 2: rho2 and v2 should appear on the sketch

Line 84 to 88: the exact definition of reflection and transmission coefficients should be given (especially for the transmission coefficient). The coefficients used here seem to be the square root of acoustic power reflection and transmission coefficient; however, if that were the case, equation (3) is incorrect, the numerator should be 2*(Z1/Z2)1/2. Here again, reference [16] is in Polish, and should be replaced by one of the numerous references in English that provide the same theoretical information (for instance Dieulesaint and Royer, ‘Elastic waves in solids, free and guide propagation’, Springer 2000, ISBN 3-540-65932-3, pages 40-63).

Equation (4) is incorrect; a factor 4 should appear in front of both cos2

Line 89, lambda is the wavelength of the longitudinal wave propagating in the plate

Figure 99, the abscissa caption should be g/lambda, not lambda/g.

Line 105, ‘the direction of the signal’ should be replaced with ‘the direction of the wave’

Section 2, exact types of transmitter and hydrophone should be given. What type of transmitter is used (piezo-ceramic?)? What is its frequency bandwidth (and that of the hydrophone)? What are the diameters of the transmitter and receiver? It could affect the frequency measurements.

Line 135, the sentence is ambiguous: not only one plate with stepwise variable thickness was used, but rather several plates with thickness equal to 2,4 …mm respectively.

Figure 5, for clarity distance ‘l’ should appear in the drawing

Section 5, more details about the parameters used in the simulation should be given. Explain what kind of simulation code OnScale is (Finite Element Analysis…), how the mesh was set, what equations were solved…How is the source modelled? (stress or displacement source? Single point over an extended area? If the transmitter is a piezo-electric one, is the piezo-electricity response modelled?).

Table 2, the percentage is computed relatively to what? The incident amplitude?

Line 191, there are English mistakes in the sentence.

Section 4.1.1: an analysis of the result should be given. The variation of the amplitude of recorded signals with the position of the plate depends on many other parameters, such as the divergence of the transmitted beam, that depends itself on the transmitter geometry (diameter, shape) relatively to the wavelength in water. Michaud et al (reference 20 of this paper) found a different trend.

Figure 9b, contrarily to what is written line 229-230, the reflection coefficient is expressed with respect to g/lambda, not g.

 Line  253, where does the weakness coefficient b come from? Was it computed from experiments or from the simulations? What is the uncertainty? Figure 11, experimental and simulated points should be added to the figure.

Figure 12, the colorbar caption is too small and cannot be read.

Line 276, ‘summation’ should read ‘simulation’

Figure 13 a): the nature of each wavepacket should be added (first incident packet, packet reflected on barrier, packet transmitted-reflected on the tank wall-transmitted back).

Figure 13 b) and c): the caption is unclear. ‘amplitudes of signals inside the model’ means probably ‘maximum pressure simulated with Onscale’ (or is it displacement? Or particle velocity?). Information about what the blue and black curves are should be added.

Section 4 (‘conclusions’): as already mentioned above, the first point is not new. Point 2 (line 307) is not new either, as the theoretical background has already been compared with experiments in other fields of research. The conclusion of line 308 is incorrect, as it is true only in this particular case and depend on many factors.

Reviewer 2 Report

The authors took up the solution at first look rather simple from a scientific point of view, but very important for practical needs, the task of defining by ultrasound a geometry of a cavern with the help of a probe, which is located inside it, in case when in the path of waves there is a metal plate of a certain thickness. Authors used both the experiment and computer simulation. They discovered "that the wavelength of the incident ultrasonic signals should be equal to half the thickness of the metal plate or an integer times smaller than this thickness." And they succeeded to find that "an ultrasonic signal with linear frequency modulation (LFM) should be used" when the thickness of the plate is unknown. It seems that such results could be easily explained on the base of resonant properties of the barrier. In this connection I incline to suggest some recommendations:

  1. It would be very useful to demonstrate shortly on the base of physical nature concerning the penetration of ultrasound through the metal plate of known thickness why it provides the chosen frequency for optimal transmitting in connection also with a quality of the surface of the metal. I think that short enough explanations would be very useful and could increase the significance of the work.
  2. It is unclear why the frequency analysis of both the reflected and the transmitted waves are carried out if without nonlinearities these values must coincide.
  3. The Fig.7 is heavily overloaded. In particular, figures e) - f) and b) - c) almost repeat each other. One pair might be omitted, and the coincidence of other couple with it could be announced by some remark in the text.

This paper is well enough written to understand main results. The manuscript seems to be suitable for publication. I am therefore convinced that such a work corresponds to the content of the Journal Acoustics and can be published there.

Round 2

Reviewer 1 Report

As I already explained in the first review, the paper presents experiments and simulations of a very simplified laboratory configuration, where the pipe is modelled as a plate immersed in water, and ultrasonic wave reflection and transmission in such a configuration has been known for decades, not only theoretically but also experimentally. The results presented here are thus not new. The revised version does not provide additional experiments as suggested in the first review, and I am afraid I cannot consider it fit for publishing in Acoustics in its present state.

Here are a few comments on the modifications made in the revised version :

The modified parts need extensive editing of English langage and style.

In the introduction, the authors state that "the best methods for the determining of cavern geometry is the ultrasonic wave method". However, this is supported by citations of papers not available to the public as they are not written in English. The authors added a few lines about the existence of other methods without any matching references.

The description of simulations with OnScale is still lacking important information about the modelled physics. For instance, how is the piezo-electricity modelled (if modelled)? With what piezoelectric constants? etc

A minimum analysis of the influential experimental parameters on the results is missing (e.g. beam width).

Item 2 of the conclusion is not accurate as the mentionned result was experimentally demonstrated in previous studies.

Author Response

Second responses

Redactor remarks

Q1: Since the measurements in the field are done in the pipe, could you use a pipe in your simulation instead of a plate? Or at least you do a 2D model where you use two plates and your Tx transducer seats between these two plates. What will  happen if you use such a configuration? Did you see ultrasonic reverberations inside the pipe? Can you solve this challenging problem?

R1: We are sorry for that in Poland the problem concerning with pipe in cavern salt is very new, so we have to starting the study from the simplest problem, and the title of this paper was little changed. The reverberation of the ultrasonic wave has been investigated according with your advice. The description of this problem is presented on this version.

Q2: In cases where the thickness of the plate barrier is unknown, what is the formula to determine the thickness from ultrasonic measurements? What is the measurement error?

R2: In case where the thickness of the plate barrier is unknown, we should firstly use a incidence signals with linear modulation frequency. This problem has been described in the previous version. The error of  the found frequency depends on the frequency and close to 10 %. However the detail description about this problem we will study.

Q3: Plate surface condition (roughness, rust) could affect ultrasonic measurement. I do not think the pipe surface is clean and smooth in the field. Why not consider surface condition effect? Such work is really valuable to the industry application.

R3: Yes you are right, but one day before measurement, inside the pipe is usually stuffed and cleaned. In other hand the measurements is performed statically step by step along the borehole, so the pipe surface can be locally considered smooth.

Reviewer #1 remarks

Q1: The modified parts need extensive editing of English langage and style.

R1: The paper has been corrected by native speaker (with this response the confirmation of the native speaker is attached).

Q2: In the introduction, the authors state that "the best methods for the determining of cavern geometry is the ultrasonic wave method". However, this is supported by citations of papers not available to the public as they are not written in English. The authors added a few lines about the existence of other methods without any matching references.

R2: Sorry for that, but we tried to find articles concerning with other methods used in the salt cavern filled with different medium, where the probe is placed in borehole, but was not success. On the other hand radar or georadar methods are used on the earth surface, not in borehole.

Q3: The description of simulations with OnScale is still lacking important information about the modelled physics. For instance, how is the piezo-electricity modelled (if modelled)? With what piezoelectric constants? etc

R3: The description of simulations concerning with reflection and transmission of the ultrasonic wave in this paper have been concerned. The principal aims of the paper are focused on the determination of a geometry of underground cavern and less with the characteristics of the transducer and receiver. However in this version there are some descriptions of these elements.

Q4. A minimum analysis of the influential experimental parameters on the results is missing (e.g. beam width)

R4. The beam width was introduced, and the results of both laboratory and simulations were analyzed and the influence of the beam width was not observed, so the problem was not described.

Q5. Item 2 of the conclusion is not accurate as the mentioned result was experimentally demonstrated in previous studies.

R5. The item 2 was rewritten.

Round 3

Reviewer 1 Report

The addition of simulation in the case where emitter and transmitter are placed between two plates improves the paper.  The new title better reflects the content of the paper.

A few corrections/additions need to be made, see attached file.

Author Response

We would like to express our sincere thanks to Redactor and Reviewer for their valuable remarks and their patience with the review of our paper. The remarks have improved the precision of description of some observations. Below there are our responses to the reviewer remarks:

Q1. Line 193, the simulation parameters are still unclear. If the wave amplitude is given in Vpp, it should mean that the whole transduction process is modeled, which implies the setting of piezo-electric material for the transducer, which is not described (piezo electric constants, thickness). Similarly, line 194, an indication of the beam width is given, but is it the result of a simulation or imposed by the user ? In other finite element software, the piezoelectricity may not be taken into account, and in such cases the source is defined as either a force source or a partical velocity source. Is it the case here ?

R1. We are sorry for mistake, the phrase “The wave amplitude is 10 Vpp..” has been replace by the waves were induced by the pressure with 5 Pa of amplitude”. Some technical parameters of the Chemkop transducer were added, but regarding on the company secrecy we are not allowed to reveal all the technique parameters. The beam widths of the used transducers were estimated from the transducer geometry parameters. However for simulation the ultrasonic waves were regarded as plane waves.

Q2. An estimate of the thickness of the plate based on the frequency spectrum, along with its uncertainty, is missing in 4.2.3.

R2. The uncertainty was estimated and added

Q3. Line 367 dissipation is not solely responsible for the amplitude decrease, but also beam geometrical spreading

R3. Yes we agree with this opinion, in our simulation the wave was considered as the plane one. However in laboratory the signals were recorded for the waves emitted from the Chemkop company transducer. The problem connected with the pipe curvature and beam width can be resolved by using optical lens theory, which will be considered next.

Q4a. Line 370 if it is at least equal to times the distance’: how many times?

R4a. Sorry for this fault, there is three times, it was added in the new version.

Q4b. (three according to the conclusion, based probably on the ratio between remaining amplitude of signal SR after numerous reflections and amplitude of SRB1 at the same time)

R4b. We very apologize for not understanding exactly the above question. However at the figures 16 and 17 the amplitudes of signals SR and SRB are decreased with time, but the times between neighboring signals in both SR and SRB are comparable and depend on the distance between two plates (pipe diameter).

Q5. The conclusion drawn from section 4.2.3 is valid in 2D, but beam spreading being different in 3D, the amplitude decay may be significantly changed, leading thus to a different minimum distance of the transducer from the cavern wall.

R5. Yes, we are in agreement with this opinion, however this problem will be next precisely considered.

Round 4

Reviewer 1 Report

Regarding the fifth question and answer in the covereletter, the authors could have added the comment in the conclusion. However, the paper in the present form is sufficiently improved to be fit for publication.